# Investigating Transfer Learning in Graph Neural Networks

**Nishai Kooverjee** [1,*] **, Steven James** [1] **and Terence van Zyl** [2]

1  School of Computer Science and Applied Mathematics, University of the Witwatersrand, Johannesburg 2000, South Africa; steven.james@wits.ac.za
2  Institute for Intelligent Systems, University of Johannesburg, Johannesburg 2092, South Africa; tvanzyl@uj.ac.za
*  Correspondence: nishai.kooverjee@gmail.com

**Abstract:** Graph neural networks (GNNs) build on the success of deep learning models by extending them for use in graph spaces. Transfer learning has proven extremely successful for traditional deep learning problems, resulting in faster training and improved performance. Despite the increasing interest in GNNs and their use cases, there is little research on their transferability. This research demonstrates that transfer learning is effective with GNNs, and describes how source tasks and the choice of GNN impact the ability to learn generalisable knowledge. We perform experiments using real-world and synthetic data within the contexts of node classification and graph classification. To this end, we also provide a general methodology for transfer learning experimentation and present a novel algorithm for generating synthetic graph classification tasks. We compare the performance of GCN, GraphSAGE and GIN across both synthetic and real-world datasets. Our results demonstrate empirically that GNNs with inductive operations yield statistically significantly improved transfer. Further, we show that similarity in community structure between source and target tasks support statistically significant improvements in transfer over and above the use of only the node attributes.

**Keywords:** graph neural networks; machine learning; transfer learning; multi-task learning





## 1. Introduction and Related Work

Deep learning has achieved success in a wide variety of problems, ranging from time-series data to images and video [1]. Data from these tasks are referred to as Euclidean [2] and specialised models such as recurrent and convolutional neural networks [3–5] have been designed to leverage the properties of such data.

Despite these successes, not all problems are Euclidean. One particular class of such problems involve *graphs*, which naturally model complex real-world settings involving objects and their relationships. Recently, deep learning approaches have been extended to graph-based domains using graph neural networks (GNNs) [6], which leverage certain topological structures and properties specific to graphs [2]. Since graphs comprise entities and the relationships between them, GNNs are said to learn relational information and may have the capacity for relational reasoning [7].

One reason for the success of deep learning models is their ability to transfer previous learning to new tasks. In image classification, this transfer leads to more robust models and faster training [8–13]. Despite the importance of transfer in deep learning, there has been little insight into the nature of transferring *relational knowledge*—that is, the representations learnt by graph neural networks. There is also no comparison of the generalisability of different GNNs when evaluated on downstream task performance. This lack of insight is in part due to the lack of a model-agnostic and task-agnostic framework and standard benchmark datasets and tasks for carrying out transfer learning experiments with GNNs.

Despite transfer learning being useful in traditional deep learning, there has been little insight gained into the nature of transferring *relational knowledge*, i.e., the representations learnt by graph neural networks. There is also no comparison of the generalisability of different GNNs when evaluated on downstream task performance. This lack is partly

due to the lack of a model-agnostic and task-agnostic framework for carrying out transfer learning experiments with GNNs; and partly due to a lack of standard benchmark datasets and tasks.

We conduct an empirical study within the contexts of *node classification* and *graph classification* to determine whether transfer in GNNs occur and, if so, what factors influence success. In particular, we make the following contributions: First, we provide a methodology and additional metrics for evaluating GNN transfer learning empirically. Second, we provide a novel method for creating synthetic graph classification tasks with community structure. Finally, we evaluate the transferability of several popular GNNs on both real and synthetic datasets. Our results demonstrate that we can achieve positive transfer using graph neural networks; and that certain models exploit strong community structure properties present in the source task to yield effective transfer.

### 1.1. Background

In this work, we consider problems for which the data can be modelled as a graph. A graph $G = \{V, E\}$ consists of a set of $N$ *vertices*, $V$, and a set of *edges* $E$. Let $v_i \in V$ denote a vertex, and $e_{ij} = (v_i, v_j) \in E$ denote a directed edge from $v_i$ to $v_j$. The *adjacency matrix*, $A \in \mathbb{R}^{N \times N}$, has $a_{ij} = 1$ where $e_{ij} \in E$, and 0 otherwise. A graph may have *node attributes* of length $C$, where $X \in \mathbb{R}^{N \times C}$ is the node feature matrix, and $\mathbf{x}_i \in \mathbb{R}^C$ is the attribute vector for node $v_i$. Similarly, a graph may have an edge attribute matrix $X^e$ [6]. An important measure of a node's connectivity is its *degree*, which is the number of edges the node is connected to [14]. We denote the degree of the $i$th node as $d_i$. The degree matrix $D = \text{diag}(d_1, \ldots, d_N)$ is the diagonal matrix containing the degrees of all vertices.

#### 1.1.1. Graph Neural Networks

The success of CNNs for computer vision problems motivate the need to formulate a counterpart to the *convolution operator* for graphs [2]. Similarly to that of Euclidean signals, the problem of convolution for graphs can be tackled from either the *spatial domain* or *spectral domain*. One popular model is Graph Convolution Networks (GCNs) [15], which make use of a *spectral* graph convolution operation stacked layer-wise. We also consider two other models, GraphSAGE [16] and Graph Isomorphism Networks (GINs) [17], which utilise *spatial* graph convolutions to aggregate information from a node's neighbourhood. GINs have previously outperformed both GCN and GraphSAGE in experimental performance on social media and biological datasets [17].

These three considered GNNs fall under the category of message-passing neural networks (MPNNs) [18]. To provide comparison of the three GNNs' operations, we rewrite them as MPNN updates:

| **Model** | **Update Rule** |
|---|---|
| GCN | $h_v^{(k)} \leftarrow \sigma\left[ W^{(k)} \cdot \left( \sum_{u \in \mathcal{N}(v) \cup \{v\}} \frac{1}{\sqrt{\tilde{d}_u \tilde{d}_v}} h_u^{(k-1)} \right) \right]$ |
| GraphSAGE | $h_v^{(k)} \leftarrow \sigma\left[ W^{(k)} \cdot \left( \frac{1}{\tilde{d}_v^{\text{in}}} \sum_{u \in \mathcal{N}(v) \cup \{v\}} h_u^{(k-1)} \right) \right]$ |
| GIN | $h_v^{(k)} \leftarrow \sigma\left[ W^{(k)} \cdot \left( (1 + \epsilon^{(k)}) \cdot h_v^{(k-1)} + \sum_{u \in \mathcal{N}(v)} h_u^{(k-1)} \right) \right]$ |

where $h_v^{(k)}$ and $W^{(k)}$ are the GNN embeddings for node $v$ and the weight matrix at the layer $k$ respectively, $\sigma(\cdot)$ is a nonlinear activation function, $\mathcal{N}(v)$ is the neighbourhood of node $v$, $\tilde{d}_v$ and $\tilde{d}_v^{\text{in}}$ are the renormalised degree and in-degree of node $v$ (see Kipf and Welling [15]), and $\epsilon^{(k)}$ is GIN's central node weighting parameter.

### 1.1.2. Transfer Learning

Deep neural network and machine learning models are usually trained for a specific task from a random initialisation of their parameters. If the task or nature of the input data changes, the network must be retrained from scratch. This retraining differs from humans, who reuse past knowledge and apply it to new contexts and tasks. The ability to reuse knowledge beyond the context in which it was learnt is known as *transfer learning* [19]. Transfer with neural networks can be achieved by simply reusing a previously-trained network's weights. The transferred weights can either be used as a starting point for training a network (*fine-tuned transfer*), or used as fixed features on the target task (*frozen-weight transfer*). To formalise transfer learning, Pan and Yang [20] define the notion of *domains* and *tasks* as used in this paper.

Taylor and Stone [19] present various metrics for the evaluation of transfer learning, including *Transfer Ratio*, *Jumpstart* and *Asymptotic Performance*. The Transfer Ratio is the ratio of the total cumulative performance of the transfer learner to the base learner, while Jumpstart is the initial performance improvement by the transfer over the base learner. Asymptotic Performance refers to the improvement made in the final learnt performance in the target task. Figure 1 below describes these metrics.

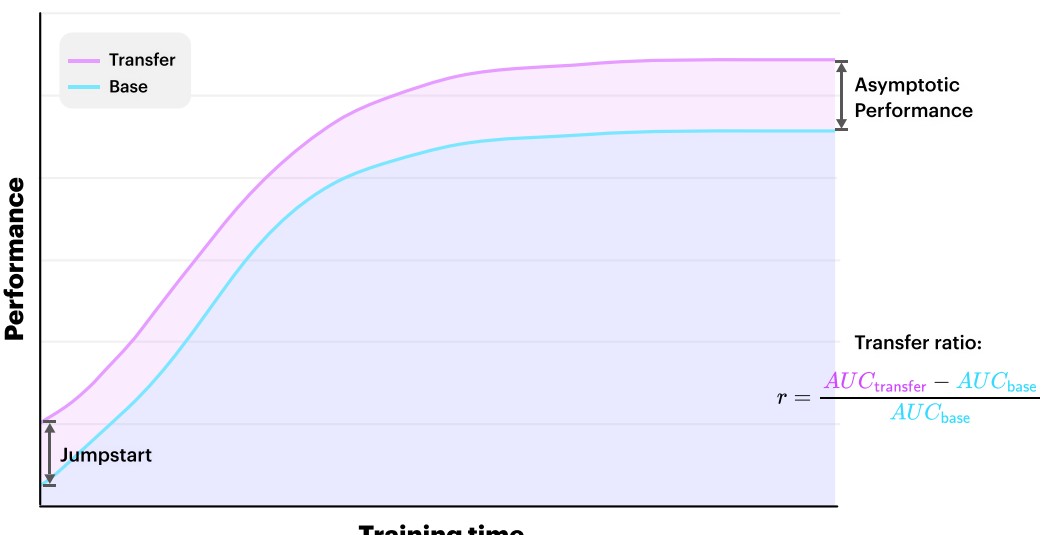

**Figure 1.** An illustration of the *jumpstart*, *asymptotic performance* and *transfer ratio* metrics. The transfer ratio is computed using the area under the curve (AUC).

### 1.1.3. Related Work

Bengio [21] notes that deep learning algorithms seem well suited to transfer learning through learning 'abstract' representations. Knowledge is represented at multiple levels, with higher level representations learnt as compositions of lower level features [22]. Kornblith et al. [23] and Huh et al. [24] investigate transfer learning with CNNs pretrained on ImageNet [25] and find that networks train faster, and achieve improved accuracy as a result.

Partly due to recency, and partly due to the multitude of approaches, not much research exists which investigates transfer learning for GNNs. Hamilton [26] notes that little success has been achieved by pretraining GNNs. This may be due to the fact that randomly initialised GNNs extract features that are just as useful as a trained network's [27].

However, Lee et al. [28] do propose a framework to transfer spectral information between source and target tasks for node classification. Experiments on real-world datasets show transfer is most effective when the source and target graphs are similar. Hu et al. [29] develop a framework to effectively pretrain GNNs for downstream tasks by pretraining a GNN at the level of individual nodes and the entire graphs. Thus, they describe two techniques to exploit node-level knowledge: context prediction and attribute masking,

as well as two approaches for graph-level pretraining. More recently, Dai et al. [30] present AdaGCN: a framework for transfer learning based on adversarial domain adaption with GCNs.

## 2. Materials and Methods

Errica et al. [31] notes that the experimental settings from GNN research papers are often ambiguous, and the results are not reproducible. Reproducibility is, therefore, a core objective, and as such, we provide the code for our experiments. We utilise the `PyTorch-Geometric` [32] library for efficient GPU-optimised implementations of the three selected GNNs [33]. We track our experiments using `Comet.ml`, and make them publicly available for transparency.

Our experiments pretrain GNNs on various source tasks and fine-tune them on a target task. All experiments are conducted in a fully supervised setting. The experiments track the performance of pretrained models as well as the randomly initialised models across training. This methodology allows us to compare transferability in terms of the transfer learning metrics described earlier. We make use of synthetic and real-world data in our experiments. To avoid the problems of a lack of meaningful data-splits [31] and the instability of training on small graph datasets [34], we make use of the Open Graph Benchmark [35] datasets for our real-world datasets. To ascertain the statistical significance of our results, we employ a pairwise two-sided *t*-test for identical means of independent sample statistics with the alternative hypothesis 'greater than' throughout.

### 2.1. Node Classification Experimental Design

Node classification involves assigning a class label to an unlabelled node in a graph. Solving this problem considers structural properties of graphs (e.g., node degree), the node's attributes, or some relationship between them. Practical applications include real-world contexts, such as social networks [36] or knowledge graphs [37]. Below we describe the datasets and experimental methodology used.

#### 2.1.1. Synthetic Data

For meaningful classifications, instances within a single class should be similar. A graph where nodes in the same class are densely connected is said to contain strong *community structure*.

The *Modularity*, defined as

$$M = \frac{1}{2|E|} \sum_{ij} \left( a_{ij} - \frac{d_i d_j}{2|E|} \right) \delta(i, j),$$

is one measure of a graph's community structure with respect to the *structure* of the network [38], where $\delta(i, j) = 1$ if $v_i$ and $v_j$ belong to the same class, and 0 otherwise.

The *Within Inertia* ratio is another measure of the community structure that takes into account node *attribute* values [39]. Given a partition of the graph's vertices $\mathcal{P}$, the Within Inertia is given by:

$$I = \frac{\sum\limits_{C \in \mathcal{P}} \sum\limits_{v \in C} \text{dist}(x_v, g_C)^2}{\sum\limits_{v \in V} \text{dist}(x_v, g)^2},$$

where $\text{dist}(x_{v_i}, x_{v_j})$ is the Euclidean distance between node attribute vectors, $g_C$ is the centre of gravity of the vertices in $C$, and $g$ the global centre of gravity of all vertices.

To generate synthetic datasets, we make use of *DANCer*: a generator for dynamic attributed networks with community structure [39]. The generator produces graphs with communities using micro-operations (local operations such as removing a node) and macro-operations (community-level operations such as splitting a community). The generator is designed with both structural and attribute homophily [40] in mind. It also models phenomena such as *preferential attachment*, where nodes are more likely to connect with

nearby or highly connected nodes [41]. These properties make DANCer an ideal generator for our purposes.

We provide four benchmark configurations: one for each combination of strong and weak structural and attribute community structure. We fix a single target task with these datasets (Configuration 4), and pretrain on the other three configurations as described in Table 1. The parentheses indicate the strength of Modularity and Within Inertia respectively—e.g., $M_\uparrow I_\downarrow$ indicates strong Modularity and weak Within Inertia.

**Table 1.** The four configurations of the synthetic node classification datasets. The average modularity and Within Inertia ratios are computed on the generated datasets.

|  | **Modularity** | | **Within Inertia** | |
|---|---|---|---|---|
| Configuration 1 ($M_\uparrow I_\uparrow$) | Strong | 0.64 | Strong | 0.37 |
| Configuration 2 ($M_\uparrow I_\downarrow$) | Strong | 0.64 | Weak | 0.47 |
| Configuration 3 ($M_\downarrow I_\uparrow$) | Weak | 0.32 | Strong | 0.39 |
| Configuration 4 ($M_\downarrow I_\downarrow$) | Weak | 0.28 | Weak | 0.99 |

### 2.1.2. Real-World Data

A common node classification domain is *citation networks*, where each node in the graph represents a publication, and edges indicates citations. We perform several transfer learning experiments using Open Graph Benchmark real-world citation networks, and complement the experiments with synthetic data. In particular, we select `Arxiv` and `MAG`—both are directed citation networks, where each node has an attribute vector containing a 128-dimensional word embedding of the paper. In addition, a paper's year of publication is also associated with its node in the network.

`Arxiv` contains 169,343 Computer Science papers, and the task is to predict which of the 40 subject areas a paper belongs to. `MAG` is taken from a subset of the Microsoft-Academic-Graph [42], and contains four types of node entities: papers, authors, institution and field of study. For consistency we will only make use of the papers, which consist of 736,389 nodes, making it a much larger and more complex network than `Arxiv`. The task here is to predict which of 349 venues (conferences or journals) each paper belongs to. Open Graph Benchmark also provides model evaluators, which use the standard *accuracy score*.

To evaluate how `MAG` transfers to itself, we split it into a *source* and a *target* graph. Papers from 2010–2014 are placed in the source split, and those from 2015–2019 belong to the target split. Any edges between nodes in separate splits are removed. Table 2 lists the statistics of the above datasets.

**Table 2.** Statistics of real-world node classification datasets used in our experiments.

|  | **Nodes** | **Edges** | **Features** | **Modularity** | **Within Inertia** | **Classes** | **Metric** |
|---|---|---|---|---|---|---|---|
| Arxiv | 169,343 | 1,166,243 | 128 | 0.495 | 0.890 | 40 | Accuracy |
| MAG (Source) | 402,598 | 1,615,644 | 128 | 0.299 | 0.813 | 349 | Accuracy |
| MAG (Target) | 333,791 | 1,390,589 | 128 | 0.286 | 0.806 | 349 | Accuracy |

### 2.1.3. Experimental Methodology

Table 3 presents the source and target tasks for our experiments using both real-world and synthetic datasets. We evaluated transfer from both `Arxiv` and the `MAG` source split to the target `MAG` split. Lastly, to investigate how important attributes are for our node classification tasks, we damaged the node attributes for both `Arxiv` and the `MAG` source graph. To damage the attributes, we replaced the attributes with Gaussian distributed random noise with a mean of 0 and a standard deviation of 1.

**Table 3.** Experiments conducted for node classification using both real-world and synthetic datasets.

|  | # | Source Task | Target Task |
|---|---|---|---|
| Real-world | 1 | Base [Random seed] | MAG (Target Split) |
|  | 2 | Arxiv |  |
|  | 3 | Arxiv [Damaged features] |  |
|  | 4 | MAG *(Source split)* [Old layer] |  |
|  | 5 | MAG *(Source split)* |  |
|  | 6 | MAG *(Source split)* [Damaged features] |  |
| Synthetic | 1 | Base [Random seed] | Configuration 4 $(M_\downarrow I_\downarrow)$ |
|  | 2 | Configuration 1 $(M_\uparrow I_\uparrow)$ |  |
|  | 3 | Configuration 2 $(M_\uparrow I_\downarrow)$ |  |
|  | 4 | Configuration 3 $(M_\downarrow I_\uparrow)$ |  |

We performed 10 runs of each of the six sets of experiments for each of the three GNNs. We used the same network architecture used by Open Graph Benchmark [35] for their experiments on `Arxiv` and `MAG`, which allows us to compare the performance for the base models as a sanity check. The network comprises three GNN layers: with an input dimensionality of 128, an output dimensionality of 349 (for `MAG`), and a hidden dimensionality of 256. We trained the networks on the target task for 2000 epochs using the Adam optimiser [43]. The best performing learning rate for each GNN was selected and fixed across our 6 experiment sets. GCN, GraphSAGE and GIN were all trained with a learning rate of 0.001 in this case.

For synthetic data experiments, we generated 10 unique graphs each for *Configurations 1*, *2*, and *3*. Throughout, we used a single instance of *Configuration 4* so that the target task was fixed. The task was 5-class node classification, and we trained the models for 2000 epochs using the Adam optimiser with a learning rate of 0.01 for GCN and GraphSAGE, and 0.001 for GIN.

*2.2. Graph Classification Experimental Design*

Graph classification is the problem of categorising whole graphs. To investigate GNN transfer for graph classification, we conducted experiments involving both real-world and synthetic problems. We followed the same experimental procedure as for node classification. We wanted to evaluate whether a dataset contains graphs with community structure at a *graph*-level. To this aim, we present an extension of the concept of community structure from node to graph level for both structural and attribute properties. The Within Inertia is a general measure of community structure since it does not depend on graph-theoretic properties, but only on the Euclidean distance between data points. Given a dataset $D$, and a partition of classes $\mathcal{P}$, a general form of the Within Inertia can be written as:

$$\frac{\sum\limits_{C \in \mathcal{P}} \sum\limits_{i \in C} \mathrm{dist}(\rho_i, g_C)^2}{\sum\limits_{i \in D} \mathrm{dist}(\rho_i, g)^2}, \tag{1}$$

where $\rho_i$ is a graph property we are interested in measuring for graph $i$ in class $C$, $g_C$ is the centre of gravity of the graphs in $C$ and $g$ is the global center of gravity of all the graphs. The Euclidean distance $\mathrm{dist}(\cdot)$ may be replaced by other distance metrics if the generation process is unknown [44,45].

We replace $\rho_i$ in Equation (1) with any property we want to measure community structure for. For the *attribute* community structure we substitute the mean of each graph's attribute matrix, $\bar{X}$, for $\rho_i$:

$$I^{\mathrm{A}} = \frac{\sum\limits_{C \in \mathcal{P}} \sum\limits_{i \in C} \mathrm{dist}\left(\bar{X}_i, g_C\right)^2}{\sum\limits_{i \in D} \mathrm{dist}\left(\bar{X}_i, g\right)^2}.$$

There is no consensus on how to approach *structural* community structure. There are numerous ways of measuring graph similarity [46], the most common being graph-edit distance [47]. Another approach is to compare graph spectra. However, these methods are slow to compute and are thus not useful for measuring entire datasets with multiple graphs.

Modularity takes node degrees as a valuable property for determining community structure for nodes. Since we wanted to measure community structure at the graph level, we used the average node degree for a graph. We substituted the average node degree, $\bar{d}_i$, for $\rho_i$ in Equation (1). This serves as the *structural* community structure measure for our datasets:

$$I^{\mathrm{S}} = \frac{\sum\limits_{C \in \mathcal{P}} \sum\limits_{i \in C} \mathrm{dist}\left(\bar{d}_i, g_C\right)^2}{\sum\limits_{i \in D} \mathrm{dist}\left(\bar{d}_i, g\right)^2}.$$

### 2.2.1. Synthetic Data

In this subsection we present a novel process for generating synthetic datasets for the task of graph classification. To create a meaningful graph classification task, we parameterise a generator to control the community structure for both structure and attributes. To do this, we generated the dataset using the following four steps:

1. Create a random *n*-class classification problem with a sample **X** and labels **y**;
2. For each label $y_i$ in **y**, generate several graphs and set their node attributes to the relevant example from **X**. Label these graphs with $y_i$;
3. Optionally, swap the labels assigned to some of the graphs to weaken community structure;
4. Optionally, replace node attributes with noise to weaken attribute community structure.

Steps 1 and 2 create community structure for attributes and structure, respectively, and the final two steps weaken the community structure if selected to do so. Figure 2 illustrates this process, where the blue blocks create community structure and the pink blocks weaken it.

The generator takes the following parameters as input:

- `num_classes`: the number of classes or labels in the dataset;
- `n_per_class`: the number of graphs to generate per class;
- `n_features`: the length of the node feature vector;
- `percent_swap`: the percentage of graphs to swap;
- `percent_damage`: the percentage of graphs where node attributes are to be damaged.

The `num_classes`, `n_per_class` and `n_features` parameters allow for dataset level properties to be varied, while `percent_swap` and `percent_damage` influence the structural and attribute community structure respectively. We fixed the size of the graphs at 30 nodes. More details regarding this generation process are provided in the Appendix A.

Similarly to the synthetic node classification datasets, we generated four configurations for each strong and weak community structure combination (see Table 4). We again indicated the strength of the Structural and Attribute Within Inertia using arrows. For example, $I_{\uparrow}^{\mathrm{S}} I_{\downarrow}^{\mathrm{A}}$ indicates strong Structural Within Inertia and weak Attribute Within Inertia. We selected *Configuration 8 ($I_{\uparrow}^{\mathrm{S}} I_{\uparrow}^{\mathrm{A}}$)* as our target task, and used the remaining three as source tasks.

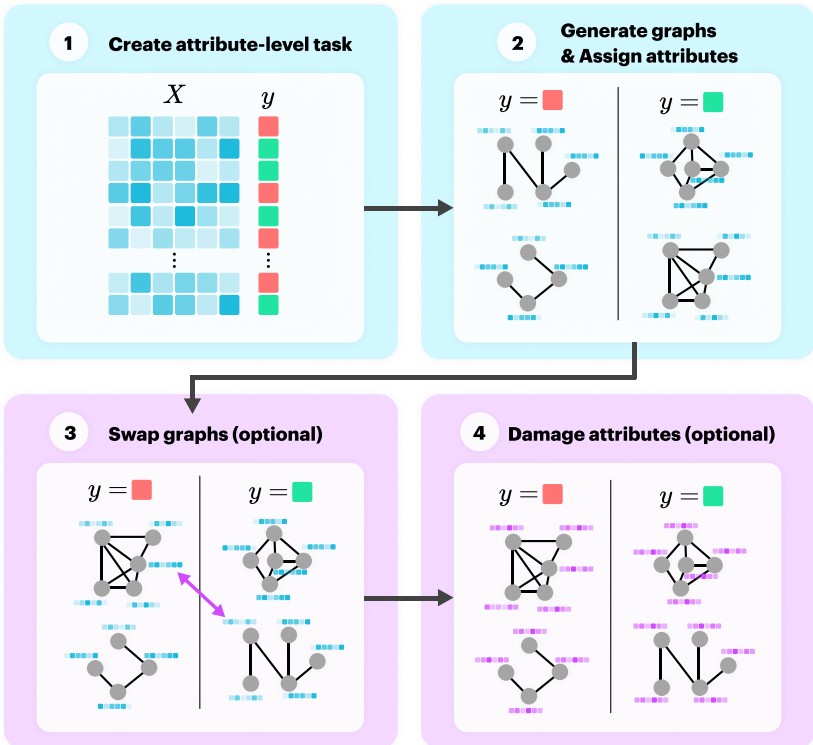

**Figure 2.** Generation process for synthetic graph classification datasets.

**Table 4.** The four configurations of synthetic graph classification datasets.

| | w.i._struct | w.i._attr | percent_swap | percent_damage |
|---|---|---|---|---|
| Configuration 5 ($I_\downarrow^S I_\downarrow^A$) | Weak | Weak | 0.95 | 0.95 |
| Configuration 6 ($I_\uparrow^S I_\downarrow^A$) | Strong | Weak | 0.92 | 0.95 |
| Configuration 7 ($I_\downarrow^S I_\uparrow^A$) | Weak | Strong | 0.95 | 0.92 |
| Configuration 8 ($I_\uparrow^S I_\uparrow^A$) | Strong | Strong | 0.92 | 0.92 |

#### 2.2.2. Real-World Data

Many real-world problems within bioinformatics and chemistry present themselves as graph classification problems. We selected datasets from Open Graph Benchmark where the task is *molecular property prediction*: BBBP and HIV. BBBP (Blood–Brain Barrier Penetration) is a physiological dataset where the task is to predict whether a given compound penetrates the blood–brain barrier or not [48]. HIV is a biophysics dataset where the task is to predict whether a compound has anti-HIV activity or not.

Both datasets are preprocessed in the same manner and are both binary classification tasks. Nodes are atoms and chemical bonds are edges, while node attributes are 9-dimensional and contain information about atomic properties. BBBP is a much smaller dataset than HIV, so we split HIV into a source and target split similar to the real-world node classification experiments: half the HIV is randomly sampled for each split, and this sample is kept fixed. A summary of the datasets is given in Table 5.

#### 2.2.3. Experimental Methodology

We conducted experiments similar to those for real-world node classification. The target task, HIV target split, is fixed, and we pretrained our GNNs on BBBP and the HIV source split and then evaluated them on the target task. We also evaluated the transfer performance where the models are pretrained on the source datasets with *damaged* node attributes, i.e., where the node attributes are replaced by Gaussian distributed random

noise with a mean of 0 and a standard deviation of 1. The experiments are described in Table 6.

**Table 5.** Statistics of real-world graph classification datasets used in our experiments.

| | No. Graphs | Average Nodes | Features | $I^S$ | $I^A$ | Classes | Metric |
|---|---|---|---|---|---|---|---|
| BBBP | 2039 | 24.06 | 9 | 0.99 | 0.98 | 2 | ROC-AUC |
| HIV (Source split) | 20,563 | 25.49 | 9 | 0.99 | 0.99 | 2 | ROC-AUC |
| HIV (Target split) | 20,564 | 25.53 | 9 | 0.99 | 0.99 | 2 | ROC-AUC |

**Table 6.** Experiments conducted for graph classification using both real-world and synthetic datasets.

| | # | Source Task | Target Task |
|---|---|---|---|
| Real-world | 1 | Base [Random seed] | HIV (Target Split) |
| | 2 | BBBP | |
| | 3 | BBBP [Damaged features] | |
| | 4 | HIV *(Source split)* | |
| | 5 | HIV *(Source split)* [Damaged features] | |
| Synthetic | 1 | Base [Random seed] | Configuration 8 ($I^S_\uparrow I^A_\uparrow$) |
| | 2 | Configuration 5 ($I^S_\downarrow I^A_\downarrow$) | |
| | 3 | Configuration 6 ($I^S_\uparrow I^A_\downarrow$) | |
| | 4 | Configuration 7 ($I^S_\downarrow I^A_\uparrow$) | |

## 3. Results

*3.1. Node Classification Results*

3.1.1. Real-World Data

In Figure 3 we see the training curves for the experiments in Table 3. We note for GCN, all the pretrained model curves rise above the base model, indicating we have positive knowledge transfer. This phenomenon is noted for both GraphSAGE (except for `Arxiv` [Damaged]), and GIN. Since the pretraining datasets are all citation networks, positive transfer is a reasonable outcome.

Concerning the Transfer Ratios in Table 7 we note that only GIN and GCN have significant transfer from the completely new `Arxiv` dataset. Interestingly, for GCN and GIN, we note that `Arxiv` with damaged attributes, in absolute terms, performs somewhat better than just `Arxiv`, indicating that graph characteristics beyond attribute values are being used for transfer. Although some of the GraphSAGE Transfer Ratios are positive, these are not statistically better than the control. Turning to Table 8, we note that GIN statistically always outperforms the other GNNs in this metric, indicating that GIN benefits the most from sharing knowledge from the source domains.

The `MAG` *(Source split)* [Old layer] tasks, in Table 7, have a greater Jumpstart than the rest, since the ouput layer does not need to be retrained. We note that both GCN and GraphSAGE show significant Jumpstart with the completely new task `Arxiv`. In fact GraphSAGE exploits graph characteristics beyond attribute values as evidenced by the `Arxiv` [Damaged] results. In Table 8 we note that GCN is either on par or significantly better than the other GNNs across all datasets on this metric; however, the result is not compelling.

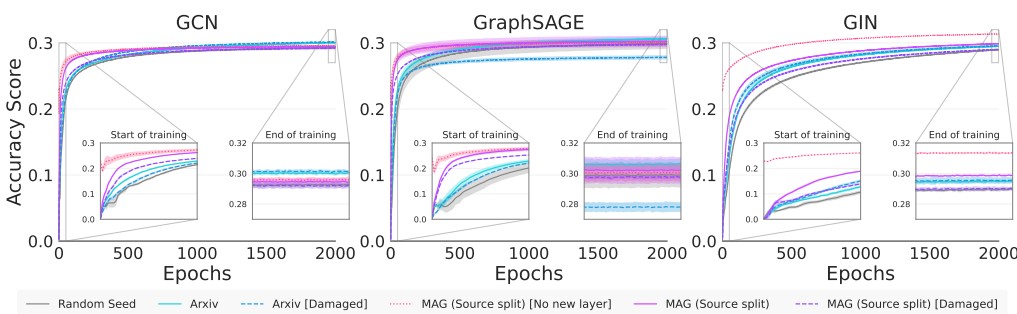

**Figure 3.** Real-world node classification training curves.

**Table 7.** Transfer metrics for real-world node classification experiments (10 runs). Bold results are positive and statistically greater than the control at $p = 0.1$. We evaluate significance for each model/metric combination.

| Model | Source Task → *MAG-Target* | Transfer Ratio | Jumpstart | Asymptotic Performance |
|---|---|---|---|---|
| Control | MAG-Source [Damaged] | $0.011 \pm 0.006$ | $0.000 \pm 0.001$ | $-0.001 \pm 0.002$ |
| **GCN** | **MAG-Source [Old layer]** | **0.042 ± 0.007** | **0.228 ± 0.023** | **0.003 ± 0.003** |
| | **MAG-Source** | **0.032 ± 0.011** | **0.001 ± 0.002** | **0.002 ± 0.003** |
| | **Arxiv** | **0.021 ± 0.007** | **0.001 ± 0.001** | **0.008 ± 0.002** |
| | **Arxiv [Damaged]** | **0.028 ± 0.006** | $0.000 \pm 0.001$ | **0.009 ± 0.002** |
| Control | MAG-Source [Damaged] | $0.023 \pm 0.037$ | $0.000 \pm 0.001$ | $0.000 \pm 0.009$ |
| **G′SAGE** | **MAG-Source [Old layer]** | $0.044 \pm 0.041$ | **0.239 ± 0.018** | $0.001 \pm 0.010$ |
| | **MAG-Source** | **0.046 ± 0.041** | $0.000 \pm 0.001$ | $0.003 \pm 0.011$ |
| | **Arxiv** | $0.028 \pm 0.040$ | **0.001 ± 0.001** | **0.007 ± 0.010** |
| | **Arxiv [Damaged]** | $-0.050 \pm 0.033$ | **0.001 ± 0.001** | $-0.021 \pm 0.009$ |
| Control | MAG-Source [Damaged] | $0.030 \pm 0.011$ | $-0.002 \pm 0.004$ | $0.000 \pm 0.002$ |
| **GIN** | **MAG-Source [Old layer]** | **0.183 ± 0.008** | **0.226 ± 0.004** | **0.024 ± 0.002** |
| | **MAG-Source** | **0.089 ± 0.007** | $-0.001 \pm 0.004$ | **0.009 ± 0.001** |
| | **Arxiv** | **0.048 ± 0.009** | $-0.001 \pm 0.004$ | **0.005 ± 0.001** |
| | **Arxiv [Damaged]** | **0.061 ± 0.016** | $-0.001 \pm 0.004$ | **0.006 ± 0.003** |

**Table 8.** Transfer metrics for real-world node classification experiments (10 runs). Bold results are not statistically greater than the best at $p = 0.1$. We evaluate significance for each source-task/metric combination.

| Source Task → *MAG-Target* (Spectral Dist.) | Model | Transfer Ratio | Jumpstart | Asymptotic Performance |
|---|---|---|---|---|
| **MAG-Source [Old layer]** | **GCN** | $0.042 \pm 0.007$ | **0.228 ± 0.023** | $0.003 \pm 0.003$ |
| | **G′SAGE** | $0.044 \pm 0.041$ | **0.239 ± 0.018** | $0.001 \pm 0.010$ |
| | **GIN** | **0.183 ± 0.008** | $0.226 \pm 0.004$ | **0.024 ± 0.002** |
| **MAG-Source** | **GCN** | $0.032 \pm 0.011$ | **0.001 ± 0.002** | $0.002 \pm 0.003$ |
| | **G′SAGE** | $0.046 \pm 0.041$ | $0.000 \pm 0.001$ | $0.003 \pm 0.011$ |
| | **GIN** | **0.089 ± 0.007** | $-0.001 \pm 0.004$ | **0.009 ± 0.001** |
| **Arxiv** | **GCN** | $0.021 \pm 0.007$ | **0.001 ± 0.001** | $0.008 \pm 0.002$ |
| | **G′SAGE** | $0.028 \pm 0.040$ | **0.001 ± 0.001** | **0.007 ± 0.010** |
| | **GIN** | **0.048 ± 0.009** | $-0.001 \pm 0.004$ | $0.005 \pm 0.001$ |

All GNNs achieve significant positive asymptotic performance above the control on the new `Arxiv` task. GraphSAGE is inconsistent and does not always show transfer at the end of training. Despite the huge Jumpstart with `MAG` *(Source split)* [Old layer], only GIN retains a large absolute improvement in Asymptotic Performance. Both GCN and GIN

once again exploit structural characteristics beyond the attribute values as evidenced by the `Arxiv` [Damaged] results. In Table 8 we see a mixture of best-performing GNNs with both GCN and GraphSAGE performing well on the completely new `Arxiv` task.

> **Takeaway 1: We have statistical evidence that transfer to a new task for node classification does occur across all metrics and GNNs. We demonstrate that GCN and GIN exploit structural rather than attribute information for achieving a positive Transfer Ratio and Asymptotic Performance, as does GraphSAGE for Jumpstart.**

Next we consider our synthetic data to interrogate exactly which characteristics of graphs are being transferred by the above GNNs.

### 3.1.2. Synthetic Data

In Figure 4 and Table 9 we note that the performance of GIN is as a result of the strong Modularity in *Configurations 1 ($M_\uparrow I_\uparrow$) and 2 ($M_\uparrow I_\downarrow$)*. For GraphSAGE, this distinction between Modularity and Within Inertia is less clear as the results for *Configurations 2 ($M_\uparrow I_\downarrow$) and 3 ($M_\downarrow I_\uparrow$)* are statistically equivalent. The absolute performance of *Configuration 3 ($M_\downarrow I_\uparrow$)* for GraphSAGE does suggest that Within Inertia is being exploited, but further investigation is required.

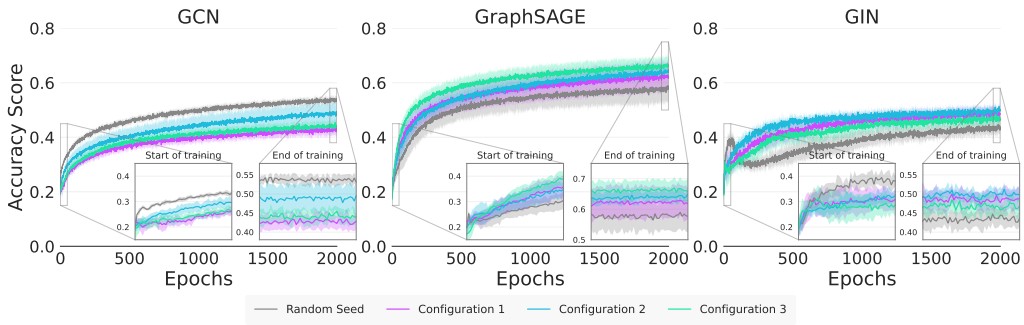

**Figure 4.** Synthetic node classification training curves.

**Table 9.** Transfer metrics for synthetic node classification (10 runs). Bold results are not statistically greater than the best at $p = 0.1$. We evaluate significance for each model/metric combination.

| Model | Source Task | Transfer Ratio | Jumpstart | Asymptotic Performance |
|---|---|---|---|---|
| **GCN** | **C.1 - $M_\uparrow I_\uparrow$** | $-0.203 \pm 0.031$ | $\mathbf{0.031 \pm 0.044}$ | $-0.103 \pm 0.026$ |
| | **C.2 - $M_\uparrow I_\downarrow$** | $-0.108 \pm 0.065$ | $\mathbf{0.012 \pm 0.039}$ | $-0.036 \pm 0.038$ |
| | **C.3 - $M_\downarrow I_\uparrow$** | $-0.176 \pm 0.026$ | $\mathbf{0.001 \pm 0.068}$ | $-0.092 \pm 0.020$ |
| **G'SAGE** | **C.1 - $M_\uparrow I_\uparrow$** | $0.083 \pm 0.086$ | $\mathbf{0.026 \pm 0.024}$ | $0.041 \pm 0.042$ |
| | **C.2 - $M_\uparrow I_\downarrow$** | $\mathbf{0.100 \pm 0.102}$ | $0.004 \pm 0.031$ | $\mathbf{0.073 \pm 0.051}$ |
| | **C.3 - $M_\downarrow I_\uparrow$** | $\mathbf{0.169 \pm 0.139}$ | $-0.024 \pm 0.042$ | $\mathbf{0.082 \pm 0.084}$ |
| **GIN** | **C.1 - $M_\uparrow I_\uparrow$** | $\mathbf{0.161 \pm 0.099}$ | $\mathbf{0.010 \pm 0.059}$ | $\mathbf{0.050 \pm 0.042}$ |
| | **C.2 - $M_\uparrow I_\downarrow$** | $\mathbf{0.211 \pm 0.076}$ | $\mathbf{0.001 \pm 0.046}$ | $\mathbf{0.061 \pm 0.029}$ |
| | **C.3 - $M_\downarrow I_\uparrow$** | $0.112 \pm 0.070$ | $\mathbf{-0.006 \pm 0.060}$ | $0.031 \pm 0.023$ |

For Jumpstart, we are unable to see any significant difference in Table 9 across all configurations for the GCN and GIN, making it unclear as to which of Modularity or Within Inertia is responsible for the positive transfer. GraphSAGE is significantly better in *Configuration 1 ($M_\uparrow I_\uparrow$)* indicating both the Strong Modularity and Within Inertia are being exploited, supporting our real-world assertion that it exploits graph characteristics beyond attribute values. We do note in Table 9 that all methods show positive Jumpstart on *Configuration 1 ($M_\uparrow I_\uparrow$) and 2 ($M_\uparrow I_\downarrow$)* suggesting that it is might be the Strong Modularity that all the GNNs are exploiting.

With Asymptotic Performance, our results do not allow us to say anything about which of structure or attributes GraphSAGE is exploiting. However, GIN, as evidenced in Table 9, is able to build on the Modularity from Configuration 1 ($M_\uparrow I_\uparrow$) as the Weak Inertia of Configuration 2 ($M_\uparrow I_\downarrow$) is not significantly better.

> **Takeaway 2: In general, GIN predominately exploits Strong Modularity for transfer—while GraphSAGE can exploit both Modularity and Within Inertia for Jumpstart—in support of our real-world data findings.**

### 3.2. Graph Classification Results

#### 3.2.1. Real-World Data

Figure 5 shows that GCN and GraphSAGE have more stable training curves than GIN, which is poorer in its performance. We observe clear transfer with GCN and GraphSAGE for both undamaged source tasks. In addition, we observe negative transfer for both damaged source tasks for GCN and GraphSAGE. GIN achieves positive self-transfer with `HIV` *(Source split)*, and negative transfer with the remaining pretrainings. GIN's training curves also show a decay over training with `BBBP` pretrainings. The training curves indicate GraphSAGE and GIN suffer from worse negative transfer than GCN.

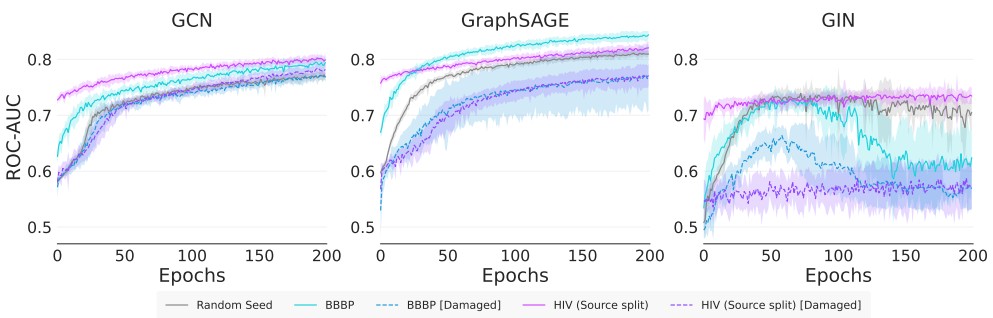

**Figure 5.** Real-world graph classification training curves.

From Table 10, we see that across the transfer metrics, GCN and GraphSAGE achieved significant positive Transfer Ratios for `HIV` *(Source split)* and `BBBP` when compared to the control. This result indicates that they are able to transfer to a completely new task. GIN shows transfer from the similar task `HIV` *(Source split)* but not from the new task `BBBP` for any of the metrics. The biggest jumpstart is seen with `HIV` *(Source split)*, which is understandable since it is a self-transfer task. None of the GNNs for any of the tasks show any significant transfer from the damaged task, indicating that the node attributes are likely exploited over graph structural characteristics.

In Table 11 we note that GraphSAGE is significantly better than all other GNNs across all metrics when transferring from the completely new `BBBP` task. For the transfer from the more similar `HIV` *(Source split)* results are mixed with GCN requiring further experimentation to determine outperformance.

> **Takeaway 3: We have significant statistical evidence to support that transfer happens for graph classification for both GCN and GraphSAGE across all metrics considered. We can also reject the hypothesis that the transfer is as a result of graph structure beyond the node attributes alone.**

In the following section, we consider synthetic data to further investigate which structural characteristics of graphs are being transferred by the GNNs.

**Table 10.** Transfer metrics for real-world graph classification experiments (10 runs). Bold results are statistically greater than the control at $p = 0.1$. We evaluate significance for each model/metric combination.

| Model | Source Task → *HIV-Target* | Transfer Ratio | Jumpstart | Asymptotic Performance |
|---|---|---|---|---|
| Control | HIV-Source [Damaged] | $-0.002 \pm 0.015$ | $0.002 \pm 0.017$ | $0.014 \pm 0.012$ |
| **GCN** | **HIV-Source** | **0.065 ± 0.010** | **0.148 ± 0.009** | **0.031 ± 0.017** |
| | **BBBP** | **0.036 ± 0.012** | **0.047 ± 0.016** | **0.026 ± 0.011** |
| | **BBBP [Damaged]** | $-0.007 \pm 0.013$ | $-0.008 \pm 0.021$ | $0.000 \pm 0.011$ |
| Control | HIV-Source [Damaged] | $-0.069 \pm 0.023$ | $-0.029 \pm 0.049$ | $-0.038 \pm 0.021$ |
| **G'SAGE** | **HIV-Source** | **0.030 ± 0.006** | **0.160 ± 0.016** | **0.011 ± 0.014** |
| | **BBBP** | **0.048 ± 0.008** | **0.072 ± 0.011** | **0.035 ± 0.009** |
| | **BBBP [Damaged]** | $-0.064 \pm 0.058$ | $-0.067 \pm 0.052$ | $-0.042 \pm 0.064$ |
| Control | HIV-Source [Damaged] | $-0.197 \pm 0.037$ | $0.039 \pm 0.048$ | $-0.130 \pm 0.051$ |
| **GIN** | **HIV-Source** | **0.033 ± 0.016** | **0.186 ± 0.045** | **0.029 ± 0.040** |
| | **BBBP** | $-0.059 \pm 0.033$ | $0.026 \pm 0.046$ | $-0.081 \pm 0.075$ |
| | **BBBP [Damaged]** | $-0.157 \pm 0.038$ | $-0.013 \pm 0.017$ | $-0.136 \pm 0.049$ |

**Table 11.** Transfer metrics for real-world graph classification experiments (10 runs). Bold results are not statistically greater than the best at $p = 0.1$. We evaluate significance for each source-task/metric combination.

| Source Task → *HIV-Target* | Model | Transfer Ratio | Jumpstart | Asymptotic Performance |
|---|---|---|---|---|
| **HIV-Source** | **GCN** | **0.065 ± 0.010** | $0.148 \pm 0.009$ | **0.031 ± 0.017** |
| | **G'SAGE** | $0.030 \pm 0.006$ | $0.160 \pm 0.016$ | $0.011 \pm 0.014$ |
| | **GIN** | $0.033 \pm 0.016$ | **0.186 ± 0.045** | **0.029 ± 0.040** |
| **BBBP** | **GCN** | $0.036 \pm 0.012$ | $0.047 \pm 0.016$ | $0.026 \pm 0.011$ |
| | **G'SAGE** | **0.048 ± 0.008** | **0.072 ± 0.011** | **0.035 ± 0.009** |
| | **GIN** | $-0.059 \pm 0.033$ | $0.026 \pm 0.046$ | $-0.081 \pm 0.075$ |

### 3.2.2. Synthetic Data

When considering the graph classification Transfer Ratios in Table 12, only Graph-SAGE and GIN showe any positive transfer. Interestingly, for both models, the result is significant for *Configuration 7 ($I_{\downarrow}^{S} I_{\uparrow}^{A}$)* indicating that it is the strong Attribute Within Inertia that is being transferred. The amount of Jumpstart achieved by all GNNs is low, as shown in Figure 6, indicating that transfer is not achieved at the start of training, and that the pretrained model performs similarly to the base models initially. For graph classification on the synthetic data, we note from Table 12 that none of the considered methods are able to achieve positive Jumpstart. These values are not discussed further here. The values for the asymptotic performance show similar results to the Transfer Ratios.

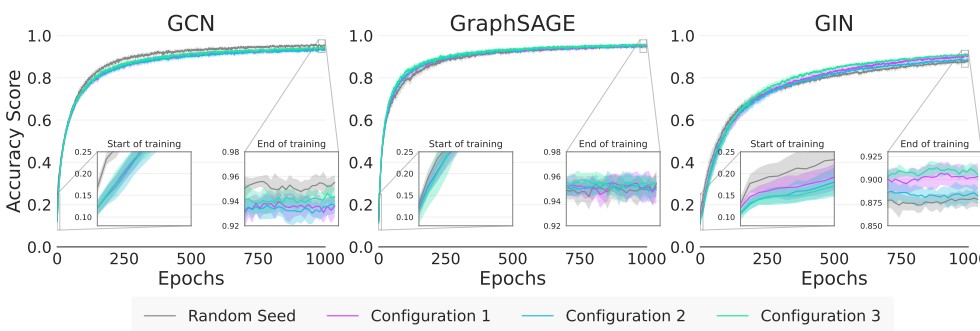

**Figure 6.** Synthetic graph classification training curves.

Since there is minimal Jumpstart, any positive or negative transfer appears to be achieved by the GNNs towards the end of training. GraphSAGE and GIN both achieve positive Asymptotic Improvements as seen Table 12; however, for GraphSAGE, these values are not significantly different from negative transfer. We note for both GraphSAGE and GIN that the best transfer occurs for *Configuration 5 ($I_\downarrow^S I_\downarrow^A$) and 7 ($I_\downarrow^S I_\uparrow^A$)*. In absolute terms, we might infer that due to the higher values in *Configuration 7 ($I_\downarrow^S I_\uparrow^A$)*, transfer results from the strong Attribute Within Inertia. However, these differences are not statistically significant and require further experimentation to confirm.

The fact that no Jumpstart but improved Asymptotic Performance is observed is interesting: it indicates that the transferred knowledge is not immediately useful but rather leads to better performance on downstream training. This observation would be supported by an argument that the transferred knowledge is not in the linear output layers but instead available in the deeper nonlinear feature layers and exposed later in training.

**Table 12.** Transfer metrics for synthetic graph classification (10 runs). Bold results are not statistically greater than the best at $p = 0.1$. We evaluate significance for each model/metric combination.

| Model | Source Task | Transfer Ratio | Jumpstart | Asymptotic Performance |
|---|---|---|---|---|
| **GCN** | **C.5 -** $I_\downarrow^S I_\downarrow^A$ | $-0.026 \pm 0.012$ | $-0.046 \pm 0.024$ | $-0.019 \pm 0.011$ |
| | **C.6 -** $I_\uparrow^S I_\downarrow^A$ | $-0.029 \pm 0.009$ | $-0.047 \pm 0.022$ | $-0.019 \pm 0.009$ |
| | **C.7 -** $I_\downarrow^S I_\uparrow^A$ | $-0.020 \pm 0.005$ | $-0.046 \pm 0.028$ | $-0.012 \pm 0.012$ |
| **G'SAGE** | **C.5 -** $I_\downarrow^S I_\downarrow^A$ | $0.007 \pm 0.006$ | $-0.009 \pm 0.030$ | $\mathbf{-0.003 \pm 0.011}$ |
| | **C.6 -** $I_\uparrow^S I_\downarrow^A$ | $0.008 \pm 0.010$ | $-0.006 \pm 0.024$ | $\mathbf{-0.005 \pm 0.014}$ |
| | **C.7 -** $I_\downarrow^S I_\uparrow^A$ | $\mathbf{0.015 \pm 0.011}$ | $-0.016 \pm 0.038$ | $\mathbf{0.001 \pm 0.011}$ |
| **GIN** | **C.5 -** $I_\downarrow^S I_\downarrow^A$ | $0.012 \pm 0.021$ | $-0.010 \pm 0.020$ | $\mathbf{0.025 \pm 0.016}$ |
| | **C.6 -** $I_\uparrow^S I_\downarrow^A$ | $0.001 \pm 0.016$ | $-0.023 \pm 0.017$ | $0.004 \pm 0.016$ |
| | **C.7 -** $I_\downarrow^S I_\uparrow^A$ | $\mathbf{0.027 \pm 0.017}$ | $-0.021 \pm 0.026$ | $\mathbf{0.027 \pm 0.013}$ |

> **Takeaway 4: There is significant evidence that GraphSAGE and GIN exploit Strong Attribute Within Inertia in order to achieve transfer. These results support our real-world findings with respect to GraphSAGE.**

## 4. Discussion

Our research presented several contributions for understanding transfer learning using graph neural networks. We proposed a framework for evaluating transfer with GNNs by testing various source tasks on a fixed target task. We employed this framework, along with transfer learning metrics and notions of community structure, to evaluate the transferability of three useful GNNs: GCN, GraphSAGE and GIN. We tested and compared these models using real-world and synthetic graph data for node-classification and graph-

classification contexts. In addition, we presented a novel procedure for generating synthetic datasets for graph classification.

All three of the GNNs we selected can transfer knowledge across training on the target task. GCN and GIN can exploit Strong Modularity in the source task, while GraphSAGE can leverage both structural and attribute information for node classification. For graph classification, it is less clear that any model exploits either attribute or structural community structure, and they appear to leverage a combination of the two to achieve transfer.

## 5. Conclusions

This research begins to standardise the procedure and metrics for evaluating transfer learning using GNNs. Research on deep learning with graphs is expanding rapidly, and understanding how we can achieve effective transfer is therefore of great benefit. There remains large scope for future research. Our results considered node and graph classification; but our experiments for transfer learning may be extended to other common graph domains such as link prediction and edge classification. Another avenue for future research is to repeat our experiments with other GNNs such as Graph Attention Network [49], and other Graph Network types described by Battaglia et al. [7].

**Author Contributions:** Conceptualization, N.K., S.J. and T.v.Z.; methodology, N.K.; software, N.K.; validation, N.K.; formal analysis, N.K.; investigation, N.K., S.J. and T.v.Z.; resources, S.J. and T.v.Z.; data curation, N.K.; writing—original draft preparation, N.K., S.J. and T.v.Z.; writing—review and editing, N.K., S.J. and T.v.Z.; visualization, N.K.; supervision, S.J. and T.v.Z.; project administration, S.J. and T.v.Z.; funding acquisition, N.K., S.J. and T.v.Z. All authors have read and agreed to the published version of the manuscript.

**Funding:** This research was funded in part by the National Research Foundation of South Africa grant number 122158.

**Data Availability Statement:** Publicly available datasets were analyzed in this study. This data can be found here: https://ogb.stanford.edu, accessed on 3 June 2021.

**Conflicts of Interest:** The authors declare no conflict of interest.

## Appendix A. Data Generation for Graph Classification

As described in the main article, we present a novel approach for synthetic graph classification datasets. We want to be able to control the level of community structure for both structure and attributes. To this end, we generated the datasets in the following four steps:

### Appendix A.1. Step 1: Create an Attribute-Level Task

The first step in the generation process is to create a attribute-level classification task, where vectors of length `n_features` are generated belonging to `num_classes` classes. The total number of these vectors is `num_classes` $\times$ `n_per_class` $\times$ 30, so that each node in each graph for each class has an attribute vector that can be assigned to it. We followed Morris et al. [50] in generating this attribute level task using the `scikit-learn` library (See https://scikit-learn.org/stable/modules/generated/sklearn.datasets.make_classification.html, accessed on 4 August 2021). This tool generates the classification task using a modified algorithm from Guyon [51]. After this step, the attributes have a high level of community structure.

### Appendix A.2. Step 2: Generate Graphs and Assign Attributes to the Graphs

Now that we have attribute vectors that are labelled, we want to generate graphs to assign them to. We wanted the graphs in each class to have different average degrees, so that strong community structure in terms of w.i.$_{struct}$ exists. A common and useful graph-generation algorithm is the *Barabási-Albert* (BA) model [52]. The BA algorithm models

preferential attachment, and takes in two parameters: the number of nodes $n$, and $m$ the number of edges to attach from a new node to existing nodes while growing the graph.

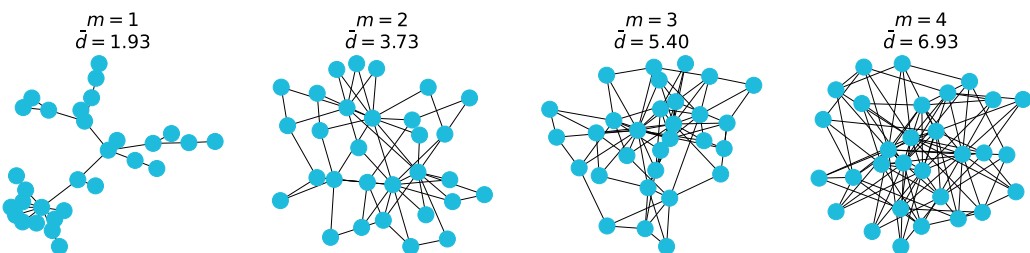

**Figure A1.** Average degrees $\bar{d}$ for varying $m$ in the Barabási-Albert model.

Varying the $m$ parameter changes the connectivity of a generated graph, and thus its average node degree. This can be seen in the figure above. By assigning graphs generated with different values of $m$ to different classes, we ensured structural community structure with respect to average node degrees.

Once `n_per_class` graphs were generated for `num_classes` with different $m$ values, the corresponding attribute vectors from the previous step were assigned to graphs with the same label. At the end of this step, we had a labelled dataset with strong community structure for both nodes and attributes.

*Appendix A.3. Step 3: Swap Graphs*

This step weakens the structural community structure of the dataset by swapping graphs. This is an optional step, and the extent to which the community structure is weakened is controlled by the `percent_swap` parameter. This parameter may be in the range $[0, 1]$, and the default value is 0 (no graphs are swapped). A random sample of pairs of graphs to swap is selected, corresponding to the specified percentage of the dataset. By swapping more graphs, the classes have less distinct average node degrees compared to one another, resulting in weaker community structure. This is demonstrated in Figure A2.

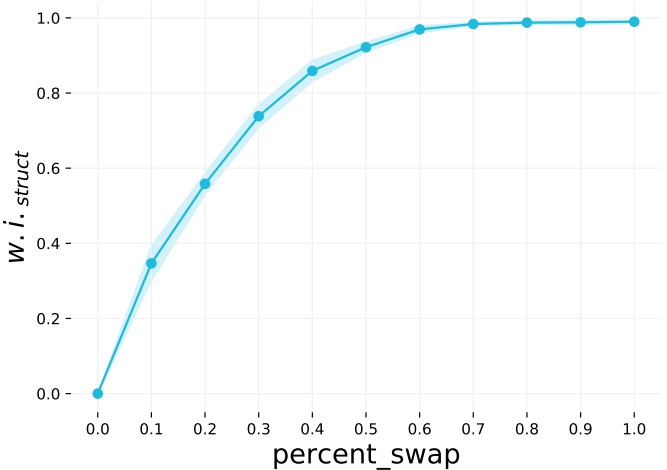

**Figure A2.** The effect of varying the `percent_swap` parameter on w.i.struct. The shaded region is the $1\sigma$ interval variance over 10 runs.

*Appendix A.4. Step 4: Damage Attributes*

The final step weakens attribute community structure by replacing node attribute vectors with random noise. This step is also optional, and is controlled by the `percent_damage` parameter. This parameter also has a range of $[0, 1]$, with a default value of 0, and defines the percentage of graphs which will have their node attributes damaged (replaced with random values). The higher the percentage is, the less distinct the attributes

from different classes are from one another, and thus a weaker community structure. This is demonstrated in Figure A3.

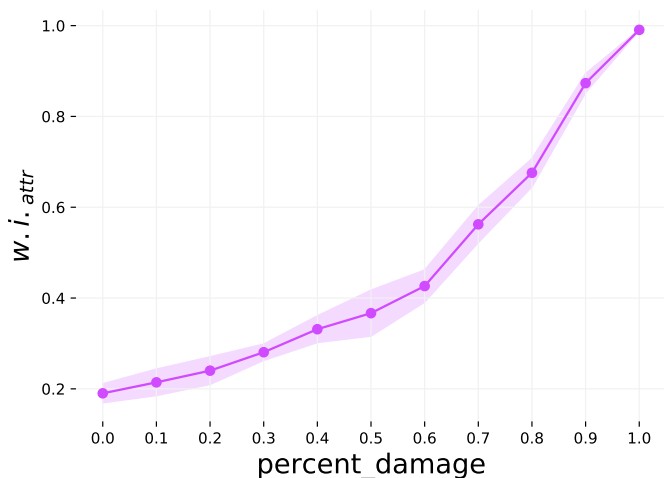

**Figure A3.** The effect of varying the `percent_damage` parameter on w.i.$_{attr}$. The shaded region is the $1\sigma$ variance over 10 runs.

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
