# Peer review of "Investigating Transfer Learning in Graph Neural Networks"

_electronics, doi:10.3390/electronics11081202_

Round 1

Reviewer 1 Report

The proposed paper is well written and scientifically founded. It has a clear structure to answer the research questions of the selected problem areas and addresses the approach in a logical and senseful way. Experiments and results are presented clearly and illustrated with helpful figures. 

In general, the authors focus on the presentation of experiments on transfer learning in graph neural networks and introduce the required mathematical background, as well as related and state-of-the-art work. The results of the experiment can serve as a foundation for future research. 

Therefore, I can strongly recommend to accept this paper - also considering this as a relevant source for further citations in related work. The only flaw is, that there is no unique approach / algorithm / modeling presented, as the paper focuses on the comparison of existing methodologies. Such a modeling can be part of future work. 

Only some minor comments:

- Line 22 / 23: duplicate usage of "involve"
- Line 28: "to transfer to new tasks" > do you mean "to being transferred to new tasks" or "to transfer <something> to new tasks"?
- Line 47: "... a set of edges E" > "... a set of C edges E" (the variable C is not introduced)
- Line 65: please consider introducing the original formulae as stated in your citations before introducing your rewritten MPNN form. 
- Line 68: "embedding" > "embeddings"
- Line 97: "note" > "notes"
- Line 149: "the authors provide" > in most of the paper, you use the "we" phrase. Therefore, this sounds a bit strange. Maybe you can rephrase this sentence
- Line 209: "We present..." > "In this subsection, we present..."

General notes: 

- it would be good to place your problem statement and / or research question more clearly somewhere in the first section. 
- i would suggest to rename the section to "Introduction and Related Work"
- the use of commas can be improved through the document. 

Reviewer 2 Report

This paper deals with the deep learning problems in graph neural networks (GNNs). The authors showed that the transfer learning is effective with GNNs. The methodology in this paper sounds feasible, and the experimental results and analyses support their argument. I have the following commends and suggestions.

  1. Line 68, what is W(k)? There are only "W" in the three equations. Which one is the typo?
  2. Line 125, the t of "t"-test should be italic. The parameters should be in italic font-style.
  3. Line 157, what is OGB? I suggest that the authors should give the full name for all the abbreviations. This will help readers understand this paper and reduce the misunderstandings. 

Reviewer 3 Report

The authors have reported the study on transfer learning in graph neural networks. The current study is interesting and novel.  The manuscript is well written and easy to follow.  In general, the main conclusions presented in the paper are well supported by the figures and supporting text. Hence, the manuscript can be accepted in its current form. However, the authors should refer to some of the recent works on transfer learning models for various applications [see  Neural Networks  136, (2021), https://doi.org/10.1016/j.neunet.2020.12.013; Neural, Comput & Applic (2022)  https://doi.org/10.1007/s00521-021-06651-x]. Hence, they should be referred. 
